# Increased Inertia Triggers Linear Responses in Motor Cortices during Large-Extent Movements—A fNIRS Study

**DOI:** 10.3390/brainsci12111539

**Published:** 2022-11-13

**Authors:** Zhi Chen, Xiaohui Song, Yongjun Qiao, Jin Yan, Chaozhe Zhu, Qing Xie, Chuanxin M. Niu

**Affiliations:** 1Department of Rehabilitation Medicine, Ruijin Hospital, Shanghai Jiao Tong University School of Medicine, Shanghai 200025, China; 2School of Medicine, Shanghai Jiao Tong University, Shanghai 200025, China; 3State Key Laboratory of Cognitive Neuroscience and Learning, Beijing Normal University, Beijing 100091, China

**Keywords:** functional near-infrared spectroscopy (fNIRS), cortical activation, motor control, voluntary movement

## Abstract

Activities of daily living consist of accurate, coordinated movements, which require the upper limbs to constantly interact with environmental loads. The magnitude of the load was shown to affect kinematic outcomes in healthy subjects. Moreover, the increase in load facilitates the recovery of motor function in patients with neurological disorders. Although Brodmann Areas 4 and 6 were found to be active during loaded movements, it remains unclear whether stronger activation can be triggered simply by increasing the load magnitude. If such a linear relationship exists, it may provide a basis for the closed-loop adjustment of treatment plans in neurorehabilitation. Fourteen healthy participants were instructed to lift their hands to their armpits. The movements were grouped in blocks of 25 s. Each block was assigned a magnitude of inertial loads, either 0 pounds (bare hand), 3 pounds, or 15 pounds. Hemodynamic fNIRS signals were recorded throughout the experiment. Both channel-wise and ROI-wise analyses found significant activations against all three magnitudes of inertia. The generalized linear model revealed significant increases in the beta coefficient of 0.001673/pound in BA4 and 0.001338/pound in BA6. The linear trend was stronger in BA6 (conditional r^2^ = 0.9218) than in BA4 (conditional r^2^ = 0.8323).

## 1. Introduction

Accurate, coordinated movements in the upper extremities are essential for activities of daily living (ADL). During ADL tasks such as eating, dressing, and self-grooming [1], the upper limbs interact constantly with loads from food, clothes, or tools. The magnitude of the load determines both the organization of muscle activation [2] and the resulting performance [3] in healthy individuals; moreover, when the magnitude is increased progressively, it facilitates the recovery of motor function in patients with neurological diseases, e.g., stroke [4] and Parkinson’s disease [5,6]. Therefore, gauging the progress of rehabilitation requires an understanding of how the brain reacts to progressively heavier loads, which is a common protocol in clinical rehabilitation [7]. On the other hand, if stronger activation in Brodmann Area 4 (BA4) and Brodmann Area 6 (BA6) can be triggered by adding loads, this potentially provides a viable path toward closed-loop, individualized therapy.

Mounting evidence has delineated the role of BA4 and BA6 for upper-limb movements against loads [8,9]. BA4 mainly contains the primary motor cortex (M1), which assumes the function of motor execution; BA6 mainly contains the supplementary motor cortex (SMA) and the premotor cortex (PMC), which assume the functions of motor planning and motor learning [10]. Experiments with fMRI have identified that the activation volume of contralateral BA4 and BA6 increases with increasing force during hand griping [11], the isometric contraction of the index finger [12], and hand squeezing [13], whereas another study claimed that only contralateral BA4 increased monotonically with force during isometric hand griping [14]. However, most fMRI studies have focused on movements of minimal joint excursions, and the load magnitude has been shown to be relatively small, partly due to the intolerance of head movements during imaging. Among other modalities of imaging, electroencephalogram (EEG) studies have found that the motor-activity-related cortical potential is linearly correlated with the isometric force output of hand grip [15] and elbow-flexion contractions [16]. Positron emission tomography (PET) and magnetoencephalography (MEG) studies reported that the activation of contralateral BA4 increased with the index-finger flexion [17] and extension force [18]. The movements in these studies were also restricted to small extents.

Functional near-infrared spectroscopy (fNIRS) is an emerging technology for brain imaging featuring high ecological validity [19], which makes it suitable for the detection of cortical responses during large-extent movements that may drag the head or trunk [20]. With regards to upper-limb movements, a recent study showed stronger activation in SMA, PMC, and M1 during voluntary movement compared to passive movement [8]. In lower-limb movements, fNIRS detected an exercise-induced decrease of deoxygenated hemoglobin in PMC [21]. Few studies have examined how mechanical load affects fNIRS responses during large-extent movements in the upper limbs.

In this study, we focused on whether increases in load inertia (i.e., the mass of an inertial load) would lead to commensurate responses in motor-related cortices within BA4 and BA6 in healthy subjects. The range, tempo, and mechanical load of the movement were chosen such that the movement was large-extent and meaningful for ADL. Our central hypothesis was that the activation of contralateral BA4 and BA6 would correlate positively with load inertia. We focused on inertial load because it is a ubiquitous type arising from interactions with daily objects, e.g., dumbbells, heavy doors, etc. [22]. This study provides an essential step toward the closed-loop adjustment of treatment plans for the rehabilitation of motor functionalities.

## 2. Materials and Methods

### 2.1. Participants

Fourteen healthy subjects participated in the study (10 males, 4 females; age = 26.5 ± 3.2; 13 right-handed, 1 mixed-handed, according to the Edinburgh handedness inventory [23]). None of the participants reported any history of neurological or musculoskeletal disorders. Before the experiment, each participant gave written consent as approved by the Ethics Committee of Ruijin Hospital, School of Medicine, Shanghai Jiao Tong University.

### 2.2. Task Design

Each subject sat on an armless chair while lifting the right hand from the hanging position to the armpit; the forearm and upper arm were required to move within the sagittal plane (Figure 1A). Subjects lifted and descended their hands repeatedly at 0.5 Hz following a metronome. Before the experiment, all subjects practiced the tasks for 3–5 min. Videos are available in Supplementary Materials for reference regarding the movement task.

The lifting movements were performed under three different load conditions: 0-pound (bare hand), 3-pound (1.36 kg), and 15-pound (6.80 kg). The 3-pound condition was chosen because it was the entry-level weight for strength training [24]; the 15-pound condition was chosen in reference to previous studies [25], and also because this was the maximum weight that allowed pilot subjects to finish the protocol without excessive fatigue.

Movements were grouped in blocks during fNIRS data collection. Each block contained 12–14 movements against a specific load magnitude. Each block lasted 25 s, and the subjects were required to rest for 40 s between adjacent blocks. The completion of all blocks took about 13 min. The sequence of the blocks was randomized. The procedure for each block was as follows:(1)The experimenter reminded the subject of the correct dumbbell for the next block;(2)The subject picked up the assigned dumbbell (if any);(3)The computer played a starting sound;(4)The subject performed lifting movements following the metronome tempo;(5)The computer played an ending sound;(6)The subject put down the dumbbell and rested.(7)Brain-hemodynamic data were acquired.

Brain-hemodynamic signals were captured using a multichannel fNIRS device (model ETG-4100, HITACHI Inc., Tokyo, Japan) at a 10-hertz sampling rate. The device evaluated the absorption of near-infrared light at two wavelengths (695 nm and 830 nm) and computed the corresponding hemoglobin and deoxyhemoglobin density following the modified Beer–Lambert Law [26]. ROIs included contralateral BA4 (mainly M1) and BA6 (encompassing SMA and PMC). BA4 was chosen because of its role in motor execution, and BA6 were chosen because of its functionality in motor planning [27]. Both BA4 and BA6 were known to be active during motor-related tasks in previous studies [28,29].

Optodes were placed on the head using a high-density 3 × 10 probe with 16 emitters and 14 detectors, which formed 44 channels covering the BA4 and BA6 of both hemispheres. The configuration of the probe is shown in Figure 1B. During the probe placement, the international 10–20 system [30] was adopted to ensure coverage of the ROIs. Specifically, we first placed the holders over subjects’ heads with channel 23 located at the CZ point, after which we set the front brim of the probe parallel to the coronal plane of the skull. After the scan, the locations of channels were measured using a 3D magnetic-space digitizer (Polhemus Patriot, Polhemus Inc., Colchester, VT, USA) to estimate the anatomical brain region of each channel. The MNI coordinates of each electrode were calculated using NFRI functions [31]. Examples of anatomic labeling and the corresponding probabilistic registrations are listed in Table 1. The flow chart of the experiment is shown in Figure 1C.

### 2.3. Brain-Hemodynamic-Signal Processing

The NIRS-KIT toolbox (version 2.0) [32] was used for brain hemodynamic signal processing. Oxy-Hb signals were used to quantify cortical activity in this study because they were sensitive to regional-cerebral-blood-flow fluctuations [33]. The pre-processing of oxy-Hb consisted of 3 steps: (1) the raw oxy-Hb signals were detrended using the linear-detrending method; (2) motion artifacts were removed using temporal derivative distribution repair (TDDR); (3) the oxy-Hb signals were band-pass filtered with cut-off frequencies at 0.01 Hz and 0.08 Hz (third-order Butterworth) to remove physiological artifacts (such as heartbeats, breath, and Mayer wave) and high-frequency noise.

The general linear model (GLM) approach [34] was used to characterize the hemodynamic response of oxy-Hb in all 44 channels under 3 conditions. Beta coefficients, of which the sign and magnitude indicate the direction (positive/negative) and intensity of change in oxy-Hb, were generated for each subject, channel, and condition. Beta coefficients are referred to interchangeably with cortical activation in this manuscript unless otherwise specified.

### 2.4. Statistical Analysis

Beta coefficients were first compared against 0 to detect cortical activation using one-sample two-tailed *t*-tests with FDR correction. Effects of weight on beta coefficients were analyzed using mixed-effect linear models, as follows:beta ~ weight+(1|subject)
where weight represents the mass of the dumbbell and the term (1|subject) accounts for subject-specific intercepts due to repeated measures. The significance level in the analyses was set at 0.05. All statistical analyses were performed using R (version 4.2.1). The models were fitted using the ‘lmer’ package [35], and the significances were calculated using the ‘lmerTest’ package [36]. The conditional r-squares were calculated using the ‘MuMIn’ package [37]; conditional r-square is the proportion of total variance explained through both fixed and random effects.

## 3. Results

We first examined the MNI coordinates and spatial registration of each channel. According to the registration records, the channels most likely to be situated in the contralateral BA4 and BA6 are marked in Figure 2. The 3D locations of all the channels are also depicted in Figure 2. The spatial registrations of all the channels for a representative subject are listed in Table 1.

### 3.1. Activation Analysis (Channel-Wise)

In the 0-pound condition (Figure 3A), significant activations were detected in five out of seven BA4 channels (7, 8, 16, 17, 18, all FDR-adjusted *p* < 0.05) as well as two out of seven BA6 channels (26, 35, all FDR-adjusted *p* < 0.05). In the 3-pound condition (Figure 3B), we discovered that all the BA4-channels (all FDR-adjusted *p* < 0.05) and six out of seven BA6 channels (24, 25, 26, 34, 35, 36, all FDR-adjusted *p* < 0.05) were significantly activated. In the 15-pound condition (Figure 3C), all the BA4 channels (all FDR-adjusted *p* < 0.05) were also significantly activated, and six out of seven BA6 channels (24, 25, 26, 34, 35, and 36, all FDR-adjusted *p* < 0.05) were activated. The overview of the activation in the ROIs map, interpolated from the averaged beta coefficients, is shown in Figure 4.

### 3.2. Activation Analysis (ROI-Wise)

To improve the spatial consistency across channels, we employed a group analysis based on ROI [38]. The three channels with the highest probability of registration were chosen for the weight adjustment of signals based on ROI. Therefore, the weight-adjusted hemodynamic signal for each ROI was:$$oxyHb_{ROI}=\frac{\sum_{i=1}^{3}P_{i}*oxy_{i}}{\sum_{i=1}^{3}P_{i}}$$
where $oxyHb_{ROI}$ represented the oxy-Hb signals of each ROI, and $P_{i}$ represented the likelihood of channel i out of three, ranked by the likelihood in the probabilistic registration.

In the 0-pound condition (Figure 5), both ROIs were significantly activated (BA4: t_13_ = 2.821, *p* < 0.05 BA6: t_13_ = 2.4619, *p* < 0.05). In the 3-pound condition, both BA4 and BA6 were significantly activated (BA4: t_13_ = 4.3567, *p* < 0.001; BA6: t_13_ = 3.7747, *p* < 0.01). In the 15-pound condition, the activation was also significant in both BA4 and BA6 (BA4: t_13_ = 4.6549, *p* < 0.001; BA6: t_13_ = 4.3725, *p* < 0.001).

### 3.3. Correlation Analysis (Channel-Wise)

We performed a traditional channel-wise analysis to test the relationship between the beta coefficients and the magnitude of load inertia. All the BA6 channels exhibited statistically significant positive correlations between load magnitudes and beta coefficients (*p* < 0.05); six out of the seven BA4 channels exhibited significant positive correlations between loads and beta coefficients (*p* < 0.05). Figure 6 shows the linear relationships from the three most significant channels. The channel-wise details of the slope, T-value, *p*-value, and location are listed in Table 2.

### 3.4. Correlation Analysis (ROI-Wise)

We ran a linear model in each ROI to examine the impact of load magnitudes on the cortical activation. As shown in Figure 7A, we discovered significantly positive correlations between the load magnitudes and the beta. The beta coefficient increased by 0.001673/pounds (conditional r^2^ = 0.8323, *p* < 0.01) in BA4. The regression lines of each subject were displayed to show individual trends. As shown in Figure 7B, the activation level of BA6 also significantly correlated with the magnitude of the load by 0.001338/pounds (conditional r^2^ = 0.9218, *p* < 0.0001). BA6 show a stronger linear trend than BA4, as suggested by the mixed-effect linear model.

## 4. Discussion

In this study, we investigated the correlation between cortical activation and the magnitude of inertial load using fNIRS. The subjects were instructed to lift their right hands to their armpits against three different loads: 0 pounds (bare hand), 3 pounds, and 15 pounds. Contralateral BA4 and BA6 activations were detected in all three conditions. Linear correlations between the beta coefficients and the magnitude of the load were identified in both channel-wise and ROI-wise analyses.

There have been debates on whether the correlation between BA4/BA6 activities and load magnitude is linear. One fMRI study claimed that only the BOLD signal in BA4, but not in BA6, increased linearly with the force output during hand gripping [14]; by contrast, another study reported that the BA6 activity showed monotonic relationships with increased force output [13]. Our results showed that the linear trends were significant in both BA4 and BA6 between the cortical activity and the load inertia. In addition, we added to existing knowledge with the observation that the linear trend seemed stronger in BA6. The difference between our results and previous reports might have been due to the larger extent of the movements in our study, i.e., larger range, higher velocity, and higher loads.

A stronger linear trend was found in BA6 (conditional r^2^ = 0.9218) than in BA4 (conditional r^2^ = 0.8323) with increased inertia. This might imply that the main effect of increasing the load inertia was to engage motor planning or learning, the suggested role of SMA and PMC within BA6 [39,40,41], rather than to engage only movement execution, which is arguably the primary role of BA4 [42]. The importance of BA6 in clinical rehabilitation has been confirmed in multiple studies [43,44]. As a result, it is interesting to ask whether a patient shows sufficient responses in the BA6 against each extra kg of load, which is a potential benchmark for clinical rehabilitation.

Our results are compatible with several studies showing that BA4 and BA6 activate more strongly against increased load during upper-limb movements. One fNIRS study reported that resistance exercise induced higher cortical activation than non-resistance exercise [9], but the load perceived by the subject was relatively small and the MNI coordinates of the activation regions were not reported. Another fNIRS study claimed that oxy-Hb exhibited an increasing trend with increasing hand-grip force [45], but the movements were restricted to a small range of motion. In general, this study provided clear data to suggest that both BA4 and BA6 linearly correlated with the magnitude of the load, sampled within the 0–15-pounds range.

In terms of clinical significance, this study might: (1) help physicians to understand how the brain reacts to progressively heavier loads, which are commonly used in clinical rehabilitation [46]; (2) provide a viable path toward the closed-loop and individualized adjustment of treatment plans; (3) provide a potential benchmark for rehabilitation assessment. Our long-term goal requires the current protocol to be generalized to patients with neurological disorders, such as stroke or traumatic brain injury. The limitations of this study include the following: (1) the patients may not be capable of lifting the arm holding the dumbbell, particularly for multiple repetitions; (2) the movement trajectories and joint angles were not motion-captured, which might be essential for the explanation of differences between healthy subjects and patients.

## 5. Conclusions

We found that during large-extent upper-limb movements, the increased magnitude of the inertial load correlated linearly with the activation in the motor-related cortices BA4 and BA6. The linear trend was stronger in BA6 than in BA4. This raises the question of whether this linear relationship is also present in patients with neurological disorders.

## Figures and Tables

**Figure 1 brainsci-12-01539-f001:**
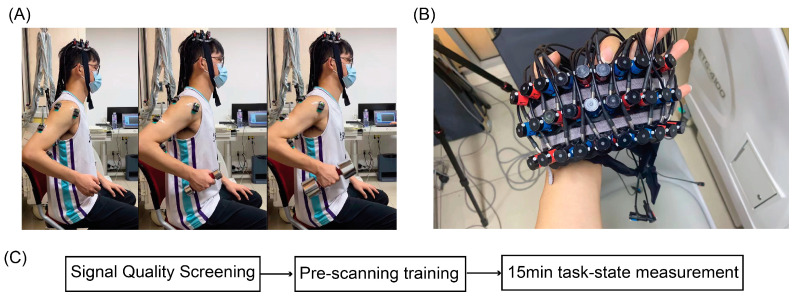
(**A**) Actual scenes of the experiment. The subject is seated on an armless chair, wearing loose clothes. (**B**) The high-density fNIRS probe. (**C**) The flow chart of experiments. Each subject passed through signal-quality screening (about 5 min), pre-scanning training (about 3 min), and task-state measurement (about 15 min).

**Figure 2 brainsci-12-01539-f002:**
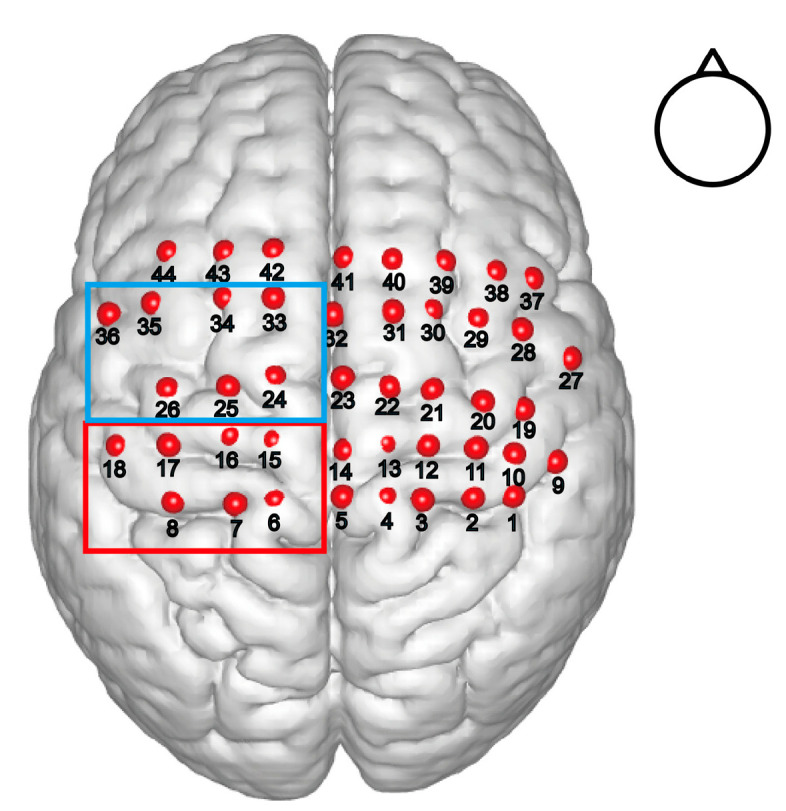
Three-dimensional location of each channel. We used NFRI fNIRS tools incorporated in NIRS-SPM to calculate the spatial registration of each channel from real space to MNI (Montreal Neurological Institute templates) coordinates and plotted them on the standard brain. We considered channels 6, 7, 8, 15, 16, 17, and 18 (red box) as the region of contralateral BA4 and channels 24, 25, 26, 33, 34, 35, and 36 (blue box) as the region of contralateral BA6 based on the probabilistic registration.

**Figure 3 brainsci-12-01539-f003:**
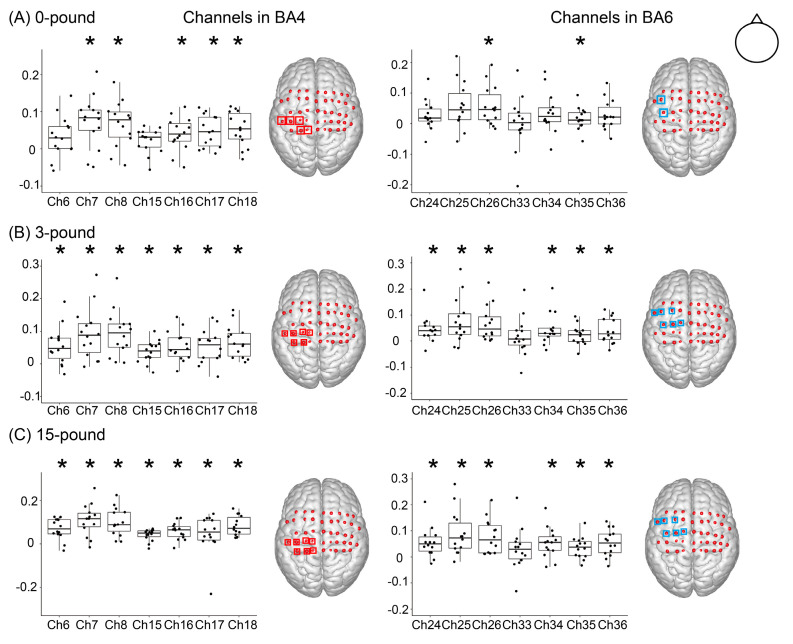
Channel-wise activation. Box-plot shows the median and percentile beta value of channels located in ROIs, and jitters showing the beta value of each subject. (**A**) The 0-pound condition. Five out of seven channels were activated in BA4 and two out of seven channels were activated in BA6. (**B**) The 3—pound condition. All channels in BA4 were activated, and six of seven channels were activated in BA6. (**C**) The 15—pound condition. The activated channels were the same as in the 3—pound condition but with a higher beta value. * FDR-adjusted *p* < 0.05 from one-sample *t*-test.

**Figure 4 brainsci-12-01539-f004:**
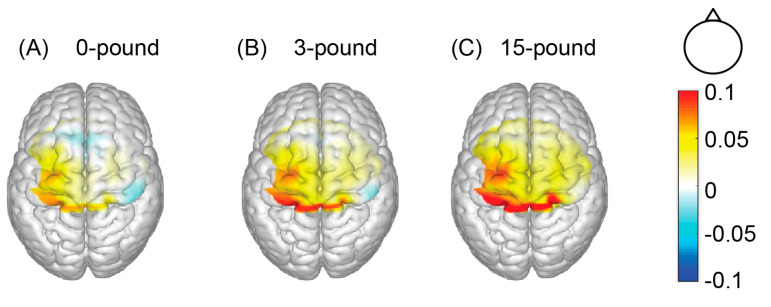
Averaged beta on a standardized brain model (n = 14). (**A**) The 0—pound condition, in which a typical contralateral activation pattern was observed. (**B**) The 3—pound condition, in which BA4 and BA6 show higher activation levels. (**C**) The 15—pound condition, in which BA4 and BA6 exhibited the highest activation among the three conditions.

**Figure 5 brainsci-12-01539-f005:**
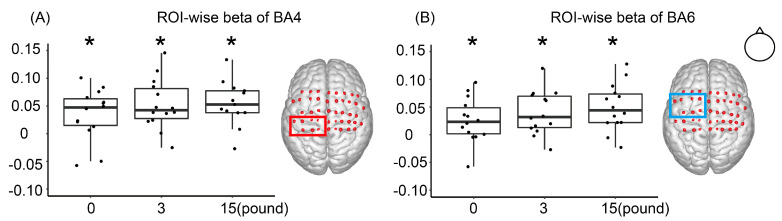
ROI-wise (Region of interest, ROI) activation. (**A**) BA4 was activated in all three conditions. (**B**) BA6 was activated in all three conditions. * *p* < 0.05, indicating beta-values were significantly greater than 0.

**Figure 6 brainsci-12-01539-f006:**
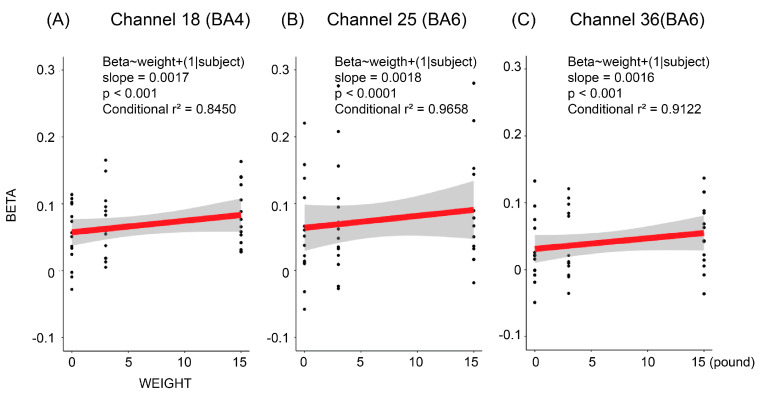
Representative examples of channel-wise correlation. (**A**) Channel 18, located in BA4. (**B**) Channel 25, located in BA6. (**C**) Channel 36, located in BA6. Representative channels were chosen based on the *p*-values of the mixed-effect model. We plotted the sample-regression line to display the overall tendency.

**Figure 7 brainsci-12-01539-f007:**
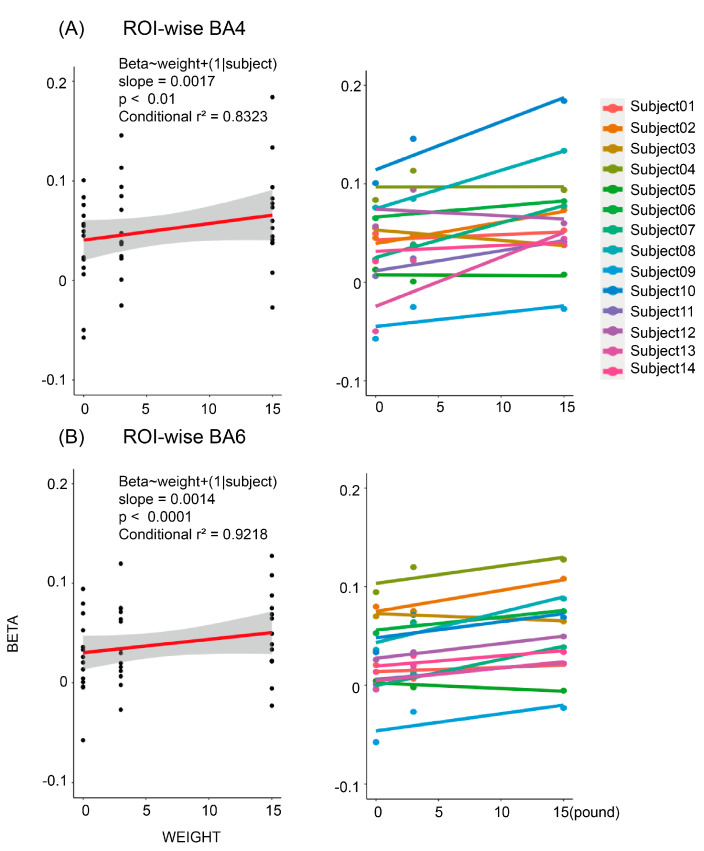
ROI-wise correlation between beta coefficients and weight. We plotted the sample-regression line to display the overall tendency. (**A**) Pooled and individual data in BA4. (**B**) Pooled and individual data in BA6. Effects of load on the cortical activation level (measured by the mean beta-value of ROIs) were fitted using mixed-effect linear models. The corresponding *p*-value and conditional r-square from mixed-effect linear models were placed above the regression line for each ROI. Significantly positive correlations were found in both ROIs.

**Table 1 brainsci-12-01539-t001:** Location of channels for subject 09.

Channel	Anatomical Label in BA	MNI Coordinate	Probability
15	4—Primary Motor Cortex	(−13.67, −31.33, 79.33)	0.868
16	4—Primary Motor Cortex	(−23.67, −31.33, 71)	0.72
24	6—Pre-Motor and Supplementary Motor Cortex	(−14.67, −14.33, 78)	0.818
25	6—Pre-Motor and Supplementary Motor Cortex	(−27.33, −16.33, 74.33)	0.853
33	6—Pre-Motor and Supplementary Motor Cortex	(−14.67, 0.33, 74)	1
34	6—Pre-Motor and Supplementary Motor Cortex	(−26.33, 0.33, 70.33)	1
35	6—Pre-Motor and Supplementary Motor Cortex	(−38.33, −1.67, 64.67)	0.982
36	6—Pre-Motor and Supplementary Motor Cortex	(−50.33, −2.67, 55.67)	0.926

BA, Brodmann Area Atlas; MNI, Montreal Neurological Institute templates.

**Table 2 brainsci-12-01539-t002:** Statistical details of channel-wise correlation.

Channel	T-Value (df = 27)	Slope	*p*-Value	Location
6	3.464	0.002129	0.0018	BA4
7	3.690	0.002125	0.0010	BA4
8	2.576	0.001797	0.0158	BA4
15	2.947	0.000988	0.0065	BA4
16	2.767	0.000995	0.0101	BA4
18	3.778	0.001724	0.0008	BA4
24	4.181	0.001505	0.0003	BA6
25	4.862	0.001784	0.0000	BA6
26	4.042	0.001224	0.0004	BA6
33	3.747	0.001739	0.0009	BA6
34	2.856	0.001007	0.0082	BA6
35	4.534	0.001260	0.0001	BA6
36	4.426	0.001585	0.0001	BA6

## Data Availability

The data presented in this study are available on request from the corresponding author.

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
