# Peer review of "Increased Inertia Triggers Linear Responses in Motor Cortices during Large-Extent Movements—A fNIRS Study"

_brainsci, 2022, doi:10.3390/brainsci12111539_

Round 1

Reviewer 1 Report

The manuscript presents the results of a single experiment with N=11 (9 males) young adults that were primarily right-handed (n-10) determined by the Edinburg Handedness Inventory.  The primary task was a dumbbell (weight) either no weight, 3 lb or 15 lb lifted in the sagittal plane 12-14 comprising a block and measurements of brain activity via a 10-Hz fNIRS system (Hitachi imager) of the prefrontal, premotor and motor cortex. Methodologically, the work is sound with respect to fNIRS signal processing and analyses as there are group-wise and ROI-wise analyses.   The figures and tables – clearly illustrate the findings and the data.   It would have been good to review the data and the analysis code.  A linear mixed effects model was used along with FDR corrections for inflation of type I error.

In addition, the following issues need to be addressed :
1)  a thorough proofing and correction of the English language to improve some grammatical and spelling errors throughout the manuscript;
2) re-analysis of the data with only the males (n =  9/11 of sample) – given that this was a small sample and that this was a strength/weight activity there may have been a floor effect for the females (n =2) for the higher weights – an alternative would be reanalysis of the data that is normalized for body weight;  
3) additional information, explanation and implications of the BA4 and BA6 is needed in the introduction and discussion of the manuscript. 

Author Response

Thank you for your careful reading, helpful comments, and constructive suggestions. We have carefully considered all comments and revised our manuscript accordingly. Please find our point-by-point responses in the attachment.

Reviewer 2 Report

The Manuscript: „ Increased inertia triggers linear responses in motor cortices during large-extent movements – an fNIRS study’’ by Zhi Chen and colleagues  analyses the consequences of increases in load inertia in sustaining commensurate responses in motor-related cortices within BA4 66 and BA6 in healthy subjects. Based on the results of the study, the authors identified a stronger linear trend of commensurate responses in motor-related cortices in BA6 than in BA4 subjects. After going through the manuscript, I have few comments for the authors:

1.     BA4 and BA6 terms need to be described more precisely in the manuscript.

2.     The number of participants in the study is small. I doubt whether the sample size was big enough to draw the statistical conclusions.

3.     Although the results and message of the study are promising, there are few limitations in addition to the ones that the authors have mentioned in the manuscript: small sample size, unequal ratio of males and females. What was the range of the subjects?

4.     Please briefly discuss the clinical significance of the study.

5.     There are few grammatical and syntax errors in the manuscript. I would suggest double checking of the manuscript to minimize grammatical flaws.

Author Response

(The authors gave the same response as above.)
